# Association between Visceral Adipose Tissue and Non-Alcoholic Steatohepatitis Histology in Patients with Known or Suspected Non-Alcoholic Fatty Liver Disease

**DOI:** 10.3390/jcm10122565

**Published:** 2021-06-10

**Authors:** Ilkay S. Idilman, Hsien Min Low, Tolga Gidener, Kenneth Philbrick, Taofic Mounajjed, Jiahui Li, Alina M. Allen, Meng Yin, Sudhakar K. Venkatesh

**Affiliations:** 1Division of Abdominal Imaging, Department of Radiology, Mayo Clinic, Rochester, MN 55905, USA; ipolater@yahoo.com (I.S.I.); hsien_min_low@ttsh.com.sg (H.M.L.); tolgagidener@gmail.com (T.G.); Philbrick.Kenneth@mayo.edu (K.P.); Li.Jiahui@mayo.edu (J.L.); Yin.Meng@mayo.edu (M.Y.); 2Department of Radiology, School of Medicine, Hacettepe University, Ankara 06100, Turkey; 3Department of Radiology, Tan Tock Seng Hospital, Singapore 308433, Singapore; 4Division of Anatomic Pathology, Mayo Clinic, Rochester, MN 55905, USA; Mounajjed.Taofic@mayo.edu; 5Division of Gastroenterology and Hepatology, Mayo Clinic, Rochester, MN 55905, USA; allen.alina@mayo.edu

**Keywords:** visceral adipose tissue, subcutaneous adipose tissue, non-alcoholic fatty liver disease, non-alcoholic steatohepatitis, hepatic fibrosis

## Abstract

(1) Purpose: To determine the association between visceral adipose tissue (VAT) and proton density fat fraction (PDFF) with magnetic resonance imaging (MRI), and hepatic steatosis (HS), non-alcoholic steatohepatitis (NASH) and hepatic fibrosis (HF) in patients with known or suspected non-alcoholic fatty liver disease (NAFLD). (2) Methods: 135 subjects that had a liver biopsy performed within 3 months (bariatric cohort) or 1 month (NAFLD cohort) of an MRI exam formed the study group. VAT volume was quantified at L2-L3 level on opposed-phase images with signal intensity-based painting using a semi-quantitative software. Liver PDFF and pancreas PDFF were calculated on fat fraction maps. Liver volume (Lvol) and spleen volume (Svol) were also calculated using a semi-automated 3D volume tool available on PACS. A histological analysis was performed by an expert hepatopathologist blinded to imaging findings. (3) Results: The mean Lvol, Svol, liver PDFF, pancreas PDFF and VAT of the study population were 2492.2 mL, 381.6 mL, 13.2%, 12.7% and 120.6 mL, respectively. VAT showed moderate correlation with liver PDFF (r = 0.41, *p* < 0.001) and weak correlation with Lvol (r = 0.38, *p* < 0.001), Svol (r = 0.20, *p* = 0.025) and pancreas PDFF (r_s_ = 0.29, *p* = 0.001). VAT, Lvol and liver PDFF were significantly higher in patients with HS (*p* < 0.001), NASH (*p* < 0.05) and HF (*p* < 0.05). VAT was also significantly higher in the presence of lobular inflammation (*p* = 0.019) and hepatocyte ballooning (*p* = 0.001). The cut-off VAT volumes for predicting HS, NASH and HF were 101.8 mL (AUC, 0.7), 111.8 mL (AUC, 0.64) and 111.6 mL (AUC, 0.66), respectively. (4) Conclusion: The MRI determined VAT can be used for predicting the presence of HS, NASH and HF in patients with known or suspected NAFLD.

## 1. Introduction

Excessive adipose tissue in the abdomen, particularly in the visceral compartment, is associated with insulin resistance, hyperglycemia, dyslipidemia, systemic hypertension, pro-thrombotic and pro-inflammatory states [1]. Adipose tissue was thought to be a passive reservoir for energy storage; however, now it is regarded as an endocrine organ as it expresses and secretes a variety of bioactive peptides, known as adipokines [2]. Furthermore, visceral adipose tissue (VAT) has a greater influence on hepatic metabolic function in contrast to subcutaneous adipose tissue (SAT) because of its direct access to the liver via the portal system [2].

Non-alcoholic fatty liver disease (NAFLD) is the leading cause of chronic liver disease with an estimated global prevalence of 24%, and is increasing worldwide [3,4]. NAFLD comprises of a spectrum ranging from simple steatosis (SS) only without any inflammation or fibrosis, and non-alcoholic steatohepatitis (NASH) which is associated with hepatocyte injury and inflammation, with or without fibrosis [5]. While there is a lower rate of progression to fibrosis in SS, it is estimated that approximately 20% of patients with NASH will develop cirrhosis in their lifetime [6]. The diagnosis of NASH is based on the histologic examination with liver biopsy which reveals hepatic steatosis (HS), ballooning of hepatocytes and lobular inflammation [5]. However, liver biopsy is invasive and has several limitations such as pain, sampling error and complications including bleeding, infection, bile leak and possible damage to other organs [7]. In view of these limitations, there is a need for a non-invasive method to detect and distinguish the progressive form, NASH, from the relatively benign course of SS.

Although body mass index (BMI) is an independent predictor of NAFLD, body fat composition, particularly visceral adipose tissue (VAT), is known to have association with HS even in non-obese individuals [8]. Magnetic resonance imaging (MRI) derived proton density fat fraction (PDFF) is a non-invasive biomarker that can evaluate hepatic steatosis and correlate with the histologic grade of fatty change [9]. VAT can be easily measured using anatomic images obtained with a computerized tomography (CT) or MRI. Previous studies have shown association between VAT measured with CT and NASH; however, the results were not conclusive [10,11,12]. The ability to predict NASH from VAT with MRI is not known. An MRI may be suitable for the evaluation of both liver PDFF and VAT at the same time, which is an advantage over CT where only abdominal fat estimation can be reliably determined. In this study, we aimed to determine the association between VAT and NAFLD features in patients with known or suspected NAFLD.

## 2. Materials and Methods

This retrospective study was approved by the institutional review board at Mayo Clinic, Rochester, MN. Two cohorts from a prospective clinical trial (NCT02565446) were included in this study. The first was the bariatric cohort which consisted of 86 obese patients with suspected NAFLD who underwent an MRI within 3 months before the surgery and an intraoperative liver biopsy. The second was the clinical cohort which included 49 patients with NAFLD and at risk of having NASH evaluated by metabolic risk factors such as diabetes mellitus, dyslipidemia, hypertension and obesity. The clinical cohort underwent an MRI within 1 month before the percutaneous liver biopsy.

Patients’ age, sex, height, body weight and BMI were obtained from the clinical notes. Obesity was defined as BMI ≥30 kg/m^2^ based on World Health Organization (WHO) criteria [13]. Insulin resistance was calculated based on fasting plasma glucose and insulin values by using the following homeostasis model assessment (HOMA) and insulin resistance (IR) method calculation: plasma glucose (mg/dL) × insulin (μg/mL)/405 in 80 patients [14].

All MRI studies were performed on standard 1.5T clinical liver MRI scanners. All the patients underwent a non-contrast enhanced MRI liver protocol. The MRI sequences acquired included: coronal T2-weighted (T2W) and non-contrast enhanced T1-weighted (T1W) sequences, In- and opposed-phase sequences, iterative decomposition of water and fat with echo asymmetry and a least-squares estimation quantification sequence (IDEAL-IQ, GE Healthcare). The images produced included water only images, fat only images, proton density fat fraction (PDFF) (which produce fat fraction maps from proton density fat fraction (PDFF) and give the option to evaluate the fat fraction of the tissue) and R2* maps. All the MRI sequences included the entire liver.

### 2.1. VAT and Subcutaneous Adipose Tissue (SAT) Estimation

A single reader measured the VAT and SAT in all subjects. As the study population was comprised of obese subjects, L2-L3 level was chosen for VAT quantification. The SAT volume was quantified at T12-L1 as the anterior abdominal wall could not be completely included within the field of view at lower levels in many subjects. Both the VAT and SAT were estimated on opposed phase images using a semi-quantitative software Radiology Informatics Laboratory Contour (RIL-Contour) [15] using signal intensity-based painting of the visceral fat by one reader (Figure 1). The opposed-phase image was chosen as the India-ink artifact between the fat and water interface and helped in the demarcation of the extent of the VAT and SAT. Modifications to the automated regions highlighted by the software for finer adjustments were performed by the reader. The VAT and SAT volumes in the single slice were generated by multiplying area with slice thickness. Both the VAT and SAT volumes were expressed in milliliters (mL). SAT quantification was not possible in 49 patients as the abdominal wall touched the magnetic bore altering the anatomy or was not completely covered within the field of view, thereby limiting the segmentation.

### 2.2. Liver and Pancreas PDFF and Liver and Spleen Volumes

The liver and pancreatic fat fraction was quantified on PDFF images generated from the IDEAL-Q sequence. For each subject, the ROIs were drawn manually in eight anatomic segments by one experienced analyst (J.L) and the mean R2* corrected PDFF was calculated by averaging the values of each ROI and weighted by the area. The pancreas PDFF measurement was also performed by placing the ROIs (>1 cm^2^) over the head, body and tail of the pancreas and calculating the mean PDFF from the ROIs. All PDFF values were expressed as percentages. One reader (T.G.), blinded to the histological results, independently calculated the liver volume (Lvol) and spleen volume (Svol) on opposed-phase images using a semi-automated 3D volume tool available on PACS (Visage Imaging GmbH, Berlin, Germany).

### 2.3. Histological Analysis

Biopsy specimens were evaluated by an expert hepatopathologist blinded to the clinical and biochemical data (T.M.). Biopsies were scored using the NASH Clinical Research Network (CRN) NAFLD Activity Score (NAS) and fibrosis staging [16]. The pathologist graded hepatic steatosis (grade 0 to 3), lobular inflammation (grade 0 to 3) and ballooning [grade 0 to 2] as per NASH CRN guidelines. The NAS score was calculated as previously described [16]. Liver fibrosis staging (stage 0 to 4) was also performed.

### 2.4. Statistical Analysis

Data were summarized as mean ±SD or median (minimum maximum) for continuous variables depending on the distributional properties of the data. Normality of the variables was tested by the Kolmogorov–Smirnov test. The Student t test or Mann–Whitney U test was used to assess differences in continuous variables between groups. A one-way ANOVA analysis was performed for comparing VAT in different hepatic steatosis stages. The degree of association between continuous and/or ordinal variables was calculated by using the Pearson correlation coefficient or Spearman’s rho. The correlation coefficient (ρ) > 0.7 was considered strong, 0.4 to 0.7 was considered as moderate and lower than 0.4 was considered weak [17]. Regression analyses were performed to assess the significance of the VAT difference in patients with HS, NASH and HF after correction for age and BMI. A receiver operating curve (ROC) analysis was performed to determine the diagnostic accuracy of VAT and other continuous variables for predicting hepatic steatosis, NASH and fibrosis. Cut-off ranges were calculated using the optimal cut-off to maximize sensitivity and specificity. For all tests, a two-tailed p value of less than 0.05 was considered as statistically significant. All statistical analyses were performed on SPSS (version 22).

## 3. Results

A total of 135 patients (M/F, 94/42) with a mean ±SD age of 49.2 ± 11.3 years were included in this study. The mean ±SD BMI of the patients was 42.5 ± 10.1 kg/m^2^. The mean ±SD VAT and SAT were 120.6 ± 48.6 mL and 251.6 ± 73.7 mL, respectively. The mean ±SD liver PDFF and pancreas PDFF were 13.2 ± 8.0% and 12.7 ± 9.9%, respectively. The mean ±SD Lvol and Svol were 2492.5 ± 701.5 mL and 381.6 ± 184.4 mL, respectively.

### 3.1. Associations among MRI Parameters

The VAT showed moderate correlation with liver PDFF (r = 0.41, *p* < 0.001) and weak correlation with Lvol (r = 0.38, *p* < 0.001), Svol (r = 0.20, *p* = 0.025) and pancreas PDFF (r_s_ = 0.29, *p* = 0.001) but no correlation with BMI. The SAT showed strong correlation with BMI (r = 0.78, *p* < 0.001), a moderate correlation with Lvol (r = 0.46, *p* < 0.001) and a weak correlation with Svol (r = 0.31, *p* = 0.004) but no significant correlation with liver PDFF and pancreas PDFF (Table 1). There was also no correlation between the VAT and SAT (r = −0.04, *p* = 0.692).

There was moderate correlation between liver PDFF and Lvol (r = 0.61, *p* < 0.001) and between liver PDFF and pancreas PDFF (r_s_ = 0.43, *p* < 0.001). There was weak but significant correlation between liver PDFF and Svol (r = 0.21, *p* = 0.05). The correlations between pancreas PDFF and BMI (r_s_ = 0.24, *p* = 0.005), Lvol (r_s_ = 0.25, *p* = 0.004,) and HOMA-IR (r_s_ = 0.23, *p* = 0.037) were weak.

### 3.2. Histological Analyses

The histological analysis showed HS in 104 (77%) patients, NASH in 73 (54.1%) patients and HF in 74 (54.8%) patients. In patients with HS, the ALT (39.1 ± 28.1 U/L vs. 26.9 ± 18.3 U/L, *p* = 0.003), triglyceride (197.1 ± 95.3 vs. 121.6 ± 51.4 mg/dL, *p* < 0.001), insulin (44 ± 52 mcIU/mL vs. 17.3 ± 14.2 mcIU/mL, *p* < 0.001) and HOMA-IR (13.1 ± 18.1 vs. 3.9 ± 3.5, *p* < 0.001) levels were significantly higher in comparison with patients without HS whereas HDL (39 ± 8.4 mg/dL vs. 50.6 ± 14 mg/dL, *p* = 0.001) levels were significantly lower.

### 3.3. Associations between MRI Parameters and HS, NASH and HF

The VAT was significantly higher in patients with HS, lobular inflammation, hepatocyte ballooning, NASH and HF (Table 2). Regression analyses demonstrated statistically significant differences in VAT for HS (*p* < 0.001), NASH (*p* = 0.008) and HF (*p* < 0.001) after correction for age and BMI. VAT showed weak correlation with the steatosis grade (r = 0.31, *p* < 0.001), NAS score (r = 0.28, *p* = 0.001) and fibrosis stage (r = 0.26, *p* = 0.003). The differences in VAT according to the fibrosis stage was also significant (106.6 ± 46.3 mL, 131.2 ± 55 mL, 142 ± 48.4 mL, 138.9 ± 45.1 and 111.2 ± 23.8 mL for F0, F1, F2, F3 and F4, respectively, *p* = 0.012).

With the ROC analysis, the cut-off volumes of VAT were 101.8 mL (AUC, 0.71) for detection of HS, 111.8 mL (AUC 0.64) for detection of NASH and 111.6 mL (AUC 0.66) (Table 3) for detection of HF.

Both Lvol and PDFF were significantly higher in patients with HS, NASH and HF (Table 2). Lvol and liver PDFF also showed significant correlation with the steatosis grade (r = 0.31, *p* < 0.001 and r = 0.82, *p* < 0.001), NAS score (r = 0.27, *p* = 0.001 and r = 0.71, *p* < 0.001) and fibrosis stage (r = 0.22, *p* = 0.011 and r = 0.46, *p* < 0.001, respectively). Svol was only significantly higher in patients with hepatic fibrosis (Table 2). Svol also had statistically significant correlation with the fibrosis stage (r = 0.25, *p* = 0.003).

There was no significant difference in the SAT in patients with HS, lobular inflammation, hepatocyte ballooning, NASH and HF (Table 2). The SAT did not correlate with the steatosis grade, NAS score or fibrosis stage (Table 4). Interestingly, BMI was lower in patients with HS (*p* = 0.04) and HF (*p* = 0.01) with statistical significance (Table 2). There were significant but weak and inverse correlations between the BMI and NAS score (r = −0.25, *p* = 0.003) and fibrosis stage (r = −0.42, *p* < 0.001, Table 4).

Pancreas PDFF showed a trend to be higher in patients with HS and NASH; however, it was not statistically significant (13.3 vs. 10.6% and 13.9% vs. 11.4%, respectively). There was a weak but significant correlation between pancreas PDFF and steatosis grade (r = 0.23, *p* = 0.007). However, there was no correlation between pancreas PDFF and the NAS score and fibrosis stage. There was also no correlation with pancreas PDFF and age (r_s_ = 0.11, *p* = 0.21).

## 4. Discussion

We demonstrated that the VAT has moderate correlation with liver PDFF and weak correlation with Lvol, Svol and pancreas PDFF. In addition, there was weak correlation between VAT and the histological steatosis grade, NAS score and fibrosis stage. The study also showed that VAT has moderate to good discriminative ability for HS, NASH and HF. These findings suggest that VAT has a moderate but definite association with NAFLD changes in the liver. The SAT, however, did not show similar association with NAFLD changes but correlated with BMI and liver volume. Interestingly, BMI was lower in patients with hepatic fibrosis. We also demonstrated significant correlations between liver MRI-PDFF and the hepatic steatosis grade and NAS score.

Choudhary et al. evaluated the VAT and SAT with histologic parameters in a limited number of NAFLD patients (n = 21) [10]. They observed that SAT volume correlated significantly with hepatic steatosis; however, none of the adipose tissue volumes had any correlation with other histological variables such as lobular inflammation, ballooning and fibrosis. Yu et al. also demonstrated higher VAT in patients with NAFLD, NASH and significant fibrosis in a total of 324 NAFLD patients and 132 controls, in accordance with our study results [11]. The VAT was, however, derived from single slice computed tomography (CT). Kim et al. demonstrated that a higher VAT area is longitudinally associated with a higher risk of NAFLD, and the baseline SAT area was significantly higher in the subjects who experienced regression of their NAFLD than in the subjects who did not experience regression, regardless of their baseline VAT area [12]. One of the main findings of our study was strong correlations among liver PDFF and hepatic steatosis and the NAS score. In a previous study by Wildman-Tobriner et al. that evaluated 370 patients, strong and moderate correlations were observed between liver PDFF and hepatic steatosis and the NAS score [18].

In the situation of increased insulin resistance, it is hypothesized that pancreatic beta cells produce more insulin to meet the demand which results in beta cell apoptosis and consequent increase of adipose tissues [19]. There are many studies evaluating the metabolic effects of pancreatic fat in the literature [20,21,22,23]. Patients with higher HOMA-IR had higher pancreatic and liver fat in a study by Patel et al. [20]. Idilman et al. observed higher pancreatic fat in patients with diabetes mellitus in an NAFLD population [21]. Sharma et al. demonstrated higher pancreatic fat fractions in the head and body/tail in type 2 diabetes mellitus patients compared to healthy cohorts [22]. In the present study, we also observed statistically significant but weak correlations between pancreas PDFF and HOMA-IR. However, Kuhn et al. did not observe a significant difference in pancreatic fat in patients with normal glucose tolerance, prediabetes and type 2 diabetes [23]. Patel et al. observed that the presence of hepatic fibrosis has an adverse effect on pancreatic fat and pancreatic fat content was lower in patients with histology-determined liver fibrosis than in those without fibrosis [24]. In our study population, patients with hepatic fibrosis had less pancreas PDFF but results were not statistically significant.

There are conflicting studies in the literature about the association between abdominal fat and hepatic fat content [22,25,26,27]. Sharma et al. observed statistically significant correlations among liver fat fraction, BMI and SAT [22]. Yu et al. demonstrated statistically significant negative correlations between visceral and subcutaneous fat areas and the three indices of the degree of hepatic fatty infiltration on CT including the attenuation value of liver parenchyma, attenuation difference between liver and spleen and attenuation ratio of liver and spleen with better correlation coefficients in the visceral fat area [26]. Chiyanika et al. evaluated 52 adolescents and observed statistically significant correlations between BMI and SAT and VAT with a better r value for SAT [27]. In their study, they also observed statistically significant correlations between hepatic fat content and SAT and BMI [26]. We also observed a strong correlation between SAT and BMI. However, we did not find a statistically significant correlation between liver PDFF and BMI or SAT. Chiyanika et al. observed a correlation between BMI and pancreatic fat content in accordance with our study. However, Sharma et al. did not demonstrate a correlation between BMI and pancreatic fat content. These differences in study results likely result from the different populations studied.

There are some limitations in the present study. First, we were not able to analyze the SAT in all patients as the abdominal wall could not be completely imaged within the field of view. We also used T12-L1 as the level for determination of SAT as there was anatomic distortion or exclusion of the abdominal wall in the field of view due to patients’ obesity. It would have been ideal to measure the VAT and SAT at the same level; however, as our study population was obese it was not possible. However, our findings with SAT are similar to the results reported in the previous literature. Future studies with an MRI on patients without extreme obesity would be useful to see if there are any correlations. We used L2-3 level for the VAT estimation, and this is a standard level for measurement of both VAT and SAT.

Second, there was an interval of up to 3 months between the MRI study and liver biopsy which can affect correlation between MRI findings and histology. The duration may be significant enough for some changes to occur in the pathological features including the degree of hepatic steatosis, degree of inflammation and to a lesser extent stage of fibrosis to occur and potentially affect the correlation analysis. This was unavoidable because scheduling of the MRI and surgeries were dependent on several factors. Third, we did not compare the parameters with a control group of healthy subjects. Our inclusion criteria were to include those with histological results available and it was not feasible to perform invasive liver biopsies without clinical indication. Finally, this is a cross-sectional study with no follow-up data. Future studies are required for assessing the longitudinal changes in the analyzed parameters after the surgery.

In conclusion, VAT volume is associated with NAFLD and may be useful for predicting the presence of HS, NASH and HF in patients with known or suspected NAFLD.

## Figures and Tables

**Figure 1 jcm-10-02565-f001:**
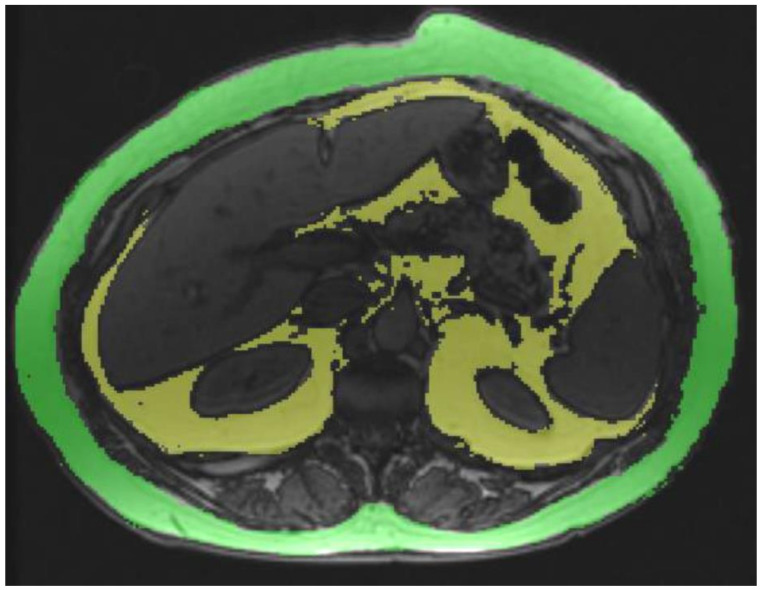
VAT area quantification at L2-L3 level on opposed-phase images using a semi-quantitative software (RIL Contour).

**Table 1 jcm-10-02565-t001:** Correlation among VAT, SAT, BMI and Lvol, Svol, liver and pancreas PDFF in the study cohort *.

	BMI	Lvol	Svol	Liver PDFF	Pancreas PDFF
VAT	r = 0.11 (*p* > 0.05)	r = 0.38 (*p* < 0.001)	r = 0.19 (*p* = 0.025)	r = 0.41 (*p* < 0.001)	r_s_ = 0.29 (*p* = 0.001)
SAT	r = 0.78 (*p* < 0.001)	r = 0.46 (*p* < 0.001)	r = 0.31 (*p* = 0.004)	r = 0.17 (*p* > 0.05)	r_s_ = 0.18, (*p* > 0.05)
BMI		r = 0.42 (*p* < 0.001)	r = 0.27 (*p* = 0.002)	r = 0.13 (*p* > 0.05)	r_s_ = 0.24 (*p* = 0.005)

Lvol, liver volume; Svol, spleen volume; VAT, visceral adipose tissue; SAT, subcutaneous adipose tissue. * = The study cohort comprised of 86 bariatric surgery subjects and 49 NAFLD subjects.

**Table 2 jcm-10-02565-t002:** Table showing differences in age, BMI, Lvol, Svol, VAT, SAT, liver PDFF and pancreas PDFF according to the presence of HS, NASH, HF, lobular inflammation and ballooning in the study cohort ^.

	Study Population	Hepatic Steatosis		*p*-Value	NASH		*p*-Value	Hepatic Fibrosis		*p*-Value	Lobular Inflammation		*p*-Value	Ballooning		*p*-Value
		Absent	Present		Absent	Present		Absent	Present		Absent	Present		Absent	Present	
Age (years)	49.2 ± 11.2	47.1 ± 10.5	49.8 ± 11.4	0.249 *	47.3 ± 10.9	50.8 ± 11.3	0.074 *	48.1 ± 11.5	50.1 ± 11	0.300 *	48.5 ± 10	49.5 ± 11.8	0.622 *	47.5 ± 10.5	50.8 ± 11.7	0.086 *
BMI (kg/m^2^)	42.5 ± 10.1	45.7 ± 10.5	41.5 ± 9.8	0.042 *	44.1 ± 9.5	41.2 ± 10.5	0.092 *	45.7 ± 8.9	39.9 ± 10.3	0.001 *	44.74 ± 10.17	41.4 ± 10	0.068 *	43.9 ± 10	41.2 ± 10.1	0.126 *
Lvol (mL)	2492.2 ± 701.5	2098.8 ± 505.9	2609.8 ± 710.7	<0.001 *	2312.4 ± 577.2	2645.4 ± 762.9	0.005 *	2285.8 ± 568.3	2662.9 ± 756.7	0.001 *	2329.25 ± 590.49	2574.1 ± 740.4	0.056 *	2304.3 ± 573.4	2672.4 ± 766.9	0.002 *
Svol (mL)	381.6 ± 184.4	329.8 ± 159.6	397.1 ± 189.2	0.074 *	357.3 ± 173	402.3 ± 192.4	0.159 *	333.9 ± 137.4	421 ± 208.5	0.004 *	355.8 ± 180.5	394.6 ± 186	0.250 *	344.7 ± 164.2	417 ± 196.7	0.022 *
VAT (mL)	120.6 ± 48.6	94.3 ± 40.8	128.5 ± 48.1	<0.001 *	109.2 ± 47.6	130.4 ± 47.5	0.011 *	106.6 ± 46.3	132.3 ± 47.6	0.002 *	106.8 ± 49.1	127.6 ± 47	0.019 *	106.8 ± 44.8	134 ± 48.6	0.001 *
SAT (mL)	251.6 ± 73.7	259.1 ± 70	248.5 ± 75.5	0.550 *	248 ± 67.3	256.6 ± 82.5	0.599 *	250 ± 68	254.4 ± 83.5	0.788 *	249.9 ± 70.2	252.9 ± 77	0.849 *	248.2 ± 70.7	256.6 ± 78.7	0.608 *
Liver PDFF (%)	13.2 ± 8	5.5 ± 2	16.3 ± 7.4	<0.001 *	9.72 ± 6.5	18 ± 7.4	<0.001 *	10.25 ± 6.7	18.11 ± 7.7	<0.001 *	9.3 ± 5.6	16.1 ± 8.4	<0.001 *	91 ± 6.7	17.8 ± 7.6	<0.001 *
Pancreas PDFF (%)	12.7 ± 9.9	10.6 ± 10	13.3 ± 9.9	0.066 **	11.4 ± 8.9	13.9 ± 10.6	0.195 **	14 ± 11.4	11.6 ± 8.4	0.322 **	11.8 ± 9.6	13.2 ± 10.1	0.403 **	12.3 ± 10.3	13.1 ± 9.7	0.350 **

Lvol, liver volume; Svol, spleen volume; VAT, visceral adipose tissue; SAT, subcutaneous adipose tissue. Student’s *t* test *. Mann–Whitney U **. ^ = The study cohort comprised of 86 bariatric surgery subjects and 49 NAFLD subjects.

**Table 3 jcm-10-02565-t003:** Diagnostic accuracy of VAT for estimation of hepatic steatosis, NASH and hepatic fibrosis in the study cohort *.

	Cut-Off Value	AUC	Sensitivity	Specificity	PPV	NPV
Hepatic Steatosis	101.8 mL	0.71 (0.60, 0.82)	0.70 (0.60, 0.78)	0.68 (0.49, 0.83)	0.88 (0.78, 0.94)	0.40 (0.27, 0.55)
NASH	111.8 mL	0.64 (0.54, 0.73)	0.61 (0.49, 0.72)	0.65 (0.51, 0.76)	0.67 (0.54, 0.78)	0.59 (0.46, 0.70)
Hepatic fibrosis	111.6 mL	0.66 (0.57, 0.76)	0.62 (0.50, 0.73)	0.64 (0.51, 0.76)	0.67 (0.55, 0.78)	0.58 (0.46, 0.70)

Numbers in parentheses are 95% confidence interval. AUC, area under the curve; NPV, negative predictive value; PPV, positive predictive value. * = The study cohort comprised of 86 bariatric surgery subjects and 49 NAFLD subjects.

**Table 4 jcm-10-02565-t004:** Correlation coefficients among steatosis grade, NAS score, fibrosis stage and BMI, Lvol, Svol, VAT, SAT, liver and pancreas PDFF in the study cohort *.

Parameter	Steatosis Grade		NAS Score		Fibrosis Stage	
	r_s_	*p*	r_s_	*p*	r_s_	*p*
BMI	−0.17	0.053	−0.25	0.003	−0.42	<0.001
Lvol	0.31	<0.001	0.27	0.001	0.22	0.011
Svol	0.14	0.100	0.13	0.127	0.25	0.003
VAT	0.31	<0.001	0.28	0.001	0.26	0.003
SAT	0.04	0.704	0.00	0.977	0.01	0.951
Liver PDFF (%)	0.82	<0.001	0.71	<0.001	0.46	<0.001
Pancreas PDFF (%)	0.23	0.007	0.17	0.058	−0.06	0.526

Lvol, liver volume; Svol, spleen volume; VAT, visceral adipose tissue; SAT, subcutaneous adipose tissue. * = The study cohort comprised of 86 bariatric surgery subjects and 49 NAFLD subjects.

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
