# Peer review of "Association between Visceral Adipose Tissue and Non-Alcoholic Steatohepatitis Histology in Patients with Known or Suspected Non-Alcoholic Fatty Liver Disease"

_jcm, 2021, doi:10.3390/jcm10122565_

Round 1

Reviewer 1 Report

The study is interesting because it is aimed at improving the non invasive diagnosis for liver disease progression from NAFLD to NASH. The main conclusion is that  VAT has a moderate but definite association with NAFLD changes in liver. Regarding the sentence"Interestingly BMI was lower in patients with hepatic fibrosis" it would be interesting the stratification of the patients for PNPLA3 genotype.  It is known that the I148M variant of PNPLA3 is involved not only in NAFLD development, but also in progression of inflammation, fibrogenesis, and carcinogenesis.

https://pubmed.ncbi.nlm.nih.gov/26409295/

Author Response

We would like to thank you for this valuable comment. Unfortunately, we don’t have the information about PNPLA3 genotype. We would consider this valuable information in our future researches.

Reviewer 2 Report

This paper investigates the relationship between pathological findings and visceral fat in patients with NAFLD or suspected NAFLD. The content itself is well examined, but it contains some major problems.

Major

  1. The authors conclude that VAT predicts steatosis, NASH, and liver fibrosis with “moderate” accuracy, but I'm afraid I have to disagree. The correlation is weak, ranging from 0.26-0.31 (Table 3). This conclusion should be revised since a moderate correlation usually indicates a correlation coefficient of 0.40-0.59.
  2. And a surprising result is included in Table 3. It is a strong correlation between Liver PDFF and steatosis/NAS (0.82/0.71).  Although liver PDFF and steatosis association have been reported, I did not know such a strong correlation of liver PDFF with NAS score. 
  3. This result should be emphasized more in this paper while investigating previous reports. The correlation coefficients of liver PDFF with ballooning and lobular inflammation, which are components of NAS other than steatosis, should also be shown.
  4. It would be perfect for measuring the PDFF of the spleen to know if it correlates with pathology. In particular, I would like to know if there is a correlation with fibrosis.
  5. This study is being examined using two different cohorts. The clinical background of each cohort should be clearly indicated in the Tables.

Minor

  1. Table 2 should be re-edited to make it easier to read.

Author Response

The authors conclude that VAT predicts steatosis, NASH, and liver fibrosis with “moderate” accuracy, but I'm afraid I have to disagree. The correlation is weak, ranging from 0.26-0.31 (Table 3). This conclusion should be revised since a moderate correlation usually indicates a correlation coefficient of 0.40-0.59.

Thank you very much for your comment. We revised the sentence as following.

“MRI determined VAT may be useful for predicting presence of HS, NASH and HF in patients with known or suspected NAFLD”

And a surprising result is included in Table 3. It is a strong correlation between Liver PDFF and steatosis/NAS (0.82/0.71).  Although liver PDFF and steatosis association have been reported, I did not know such a strong correlation of liver PDFF with NAS score.

This result should be emphasized more in this paper while investigating previous reports. The correlation coefficients of liver PDFF with ballooning and lobular inflammation, which are components of NAS other than steatosis, should also be shown.

Thank you very much for your contribution. We emphasized the high correlation among NAS score and hepatic steatosis through paper. The information about liver PDFF and ballooning and lobular inflammation is also present in table 2.

It would be perfect for measuring the PDFF of the spleen to know if it correlates with pathology. In particular, I would like to know if there is a correlation with fibrosis.

Thank you very much for your valuable contribution. Spleen does not have any significant fat deposition and therefore low PDFF values are seen. Indeed, we tried to measure spleen MRI-PDFF in another paper (you can find below) which generally gives extremely low PDFF values which is insignificant and may be just measuring the noise in the images. Therefore, we didn’t measure spleen PDFF in this research.

“Ä°dilman Ä°S, Gümrük F, HaliloÄŸlu M, Karçaaltıncaba M. The Feasibility of Magnetic Resonance Imaging for Quantification of Liver, Pancreas, Spleen, Vertebral Bone Marrow, and Renal Cortex R2* and Proton Density Fat Fraction in Transfusion-Related Iron Overload. Turk J Haematol. 2016 Mar 5;33(1):21-7. doi: 10.4274/tjh.2015.0142.”

This study is being examined using two different cohorts. The clinical background of each cohort should be clearly indicated in the Tables.

We have mentioned in the methods that the present study population was derived from two cohorts to enrich the numbers for the present study. Subjects from both cohorts are at risk of having NASH and therefore considered as a single cohort. We have added in the footnote that the subject cohort derivation on all the tables.

Minor

Table 2 should be re-edited to make it easier to read.

We increased the font size.

Reviewer 3 Report

This human study investigated the association between visceral adipose tissue and proton density fat fraction, which were measured via MRI, with the severity of NAFLD stages. Authors show that both parameters correlated with NAFLD and conclude to use these methods in prediction of NAFLD stages in patients. I have only a few recommendations.

Major:

  • The results part is very hard to read. Authors should include subheadings and show some of the key data written in the text in figures.
  • Data in the 4th results paragraph (starting with “histological analysis”) could be illustrated as figures. It is not clear, which groups were compared, when authors state “In patients with HS, there was statistically significant differences in …” because they point out two data sets but 3 patient groups.    

Minor:

  • MRI is always abbreviated and not explained. Authors should give also readers without the medical background the chance to understand the method.
  • A lot of abbreviations were used that makes reading more complicated for people, who are not familiar with all these terms. Please check to write the full name first. This was not done for MRI, CT, SAT…

Author Response

This human study investigated the association between visceral adipose tissue and proton density fat fraction, which were measured via MRI, with the severity of NAFLD stages. Authors show that both parameters correlated with NAFLD and conclude to use these methods in prediction of NAFLD stages in patients. I have only a few recommendations.

Major:

The results part is very hard to read. Authors should include subheadings and show some of the key data written in the text in figures.

Thank you very much for your recommendation. We included subheadings in the results section.

Data in the 4th results paragraph (starting with “histological analysis”) could be illustrated as figures. It is not clear, which groups were compared, when authors state “In patients with HS, there was statistically significant differences in …” because they point out two data sets but 3 patient groups.   

Thank you very much for your valuable contribution. We changed the paragraph for clarification.

Minor:

MRI is always abbreviated and not explained. Authors should give also readers without the medical background the chance to understand the method.

Thank you very much for your attention. We gave the meaning of abbreviation in the first usage. We also mentioned about MRI-PDFF in the materials and methods section.

A lot of abbreviations were used that makes reading more complicated for people, who are not familiar with all these terms. Please check to write the full name first. This was not done for MRI, CT, SAT…

Thank you very much for your attention. We checked for all abbreviations as suggested and have introduced the full expansion when used in the manuscript for the first time.

Round 2

Reviewer 2 Report

Thank you for answering my question. I will assume that appropriate improvements have been made. I wish you continued success in the future.

Reviewer 3 Report

The authors improved the manuscript in several aspects suggested by the reviewer. I recommend it now for publication.